# Optogenetic dissection of descending behavioral control in *Drosophila*

Jessica Cande[1], Shigehiro Namiki[1,2], Jirui Qiu[3], Wyatt Korff[1], Gwyneth M Card[1], Joshua W Shaevitz[4,5], David L Stern[1]*, Gordon J Berman[3,6]*

[1]Janelia Research Campus, Howard Hughes Medical Institute, Ashburn, United States; [2]Research Center for Advanced Science and Technology, University of Tokyo, Tokyo, Japan; [3]Department of Physics, Emory University, Atlanta, Georgia; [4]The Lewis-Sigler Institute for Integrative Genomics, Princeton University, Princeton, United States; [5]Department of Physics, Princeton University, Princeton, United States; [6]Department of Biology, Emory University, Atlanta, Georgia

**Abstract** In most animals, the brain makes behavioral decisions that are transmitted by descending neurons to the nerve cord circuitry that produces behaviors. In insects, only a few descending neurons have been associated with specific behaviors. To explore how descending neurons control an insect's movements, we developed a novel method to systematically assay the behavioral effects of activating individual neurons on freely behaving terrestrial *D. melanogaster*. We calculated a two-dimensional representation of the entire behavior space explored by these flies, and we associated descending neurons with specific behaviors by identifying regions of this space that were visited with increased frequency during optogenetic activation. Applying this approach across a large collection of descending neurons, we found that (1) activation of most of the descending neurons drove stereotyped behaviors, (2) in many cases multiple descending neurons activated similar behaviors, and (3) optogenetically activated behaviors were often dependent on the behavioral state prior to activation.
DOI: https://doi.org/10.7554/eLife.34275.001

*For correspondence:
sternd@janelia.hhmi.org (DLS);
gordon.berman@emory.edu (GJB)

Competing interests: The authors declare that no competing interests exist.

## Introduction

As animals navigate a dynamic environment, their survival depends on their ability to execute specific motor programs and to adjust motor output in response to external stimuli. While the brain performs computations essential for behavior, the motor circuits that directly control behavior are located close to the muscles that they control in the vertebrate spinal cord and insect ventral nerve cord. Information to drive motor patterns must therefore be transmitted from the brain to the nerve cord to direct behavior. Since there are many fewer descending neurons than neurons in the central brain, descending neurons generate an information processing bottleneck, which may generate a fundamental problem in information coding.

In flies, descending commands from the brain to the ventral nerve cord are transmitted through an estimated 250–550 pairs of descending neurons that arborize in 20 highly conserved clusters in the brain involved in sensory processing and motor behavior (*Gronenberg and Strausfeld, 1990*; *Hsu and Bhandawat, 2016*). Each descending neuron extends a single axon through the neck connective to the ventral nerve cord, where they synapse onto interneurons associated with leg, neck, and wing motor circuitry (*Namiki et al., 2018*).

Little is known about how so few neurons – approximately 0.5% of all neurons in the fly (*Alivisatos et al., 2012*) – encode signals from the brain to control the full range of movements performed by a freely moving fly. Several potential models have been suggested. One possibility is that, as with vertebrates, many stereotyped insect behaviors, such as walking, flying, or 'singing' can

be decomposed into individual motor modules controlled by central pattern generators located in the ventral nerve cord. Several recent findings in *Drosophila melanogaster*, together with earlier electrophysiological studies in larger insects, support this idea. Activation of some individually identifiable descending neurons triggers specific motor outputs, such as courtship song (*von Philipsborn et al., 2011*) backwards walking (*Bidaye et al., 2014*), or escape behavior (*King and Wyman, 1980*). However, some descending neurons modify motor programs, rather than trigger them. For example, cricket walking initiation, speed, and turning appear to depend on separately encoded descending commands (*Böhm and Schildberger, 1992*; *Gras and Kohstall, 1998*). Alternatively, motor activity may result from the summed activity of multiple descending neurons (*Heinrich, 2002*). For example, a cluster of descending neurons linking fly visual centers in the brain to the flight apparatus in the ventral nerve cord (*Strausfeld and Gronenberg, 1990*; *Namiki et al., 2018*) supports the idea that at least some descending neurons may function this way.

Descending neuron function may also be altered by behavioral state. For example, descending neuron sensory responses have been shown to be modified by locomotor state (*Staudacher and Schildberger, 1998*). This kind of modulation has been observed in other contexts, such as the effect of the neuromodulator pyrokinin on the oscillatory mechanisms underlying the crustacean gastric mill central pattern generator (*Marder, 2011*). However, it has not previously been possible to undertake a systematic analysis of the context dependency across the descending neuron population.

Systematic dissection of descending motor control is challenging for two reasons. First, it has been difficult to precisely manipulate a large number of descending neurons individually in freely behaving animals. Second, we have not had a high-throughput, unbiased behavioral phenotyping pipeline capable of objectively categorizing all of an individual's movements. Historically, insect descending neuron anatomy, connectivity and function have been described by backfilling neurons with dye and recording from individual neurons in locusts, grasshoppers, and cockroaches (for a review see [*Strausfeld et al., 1984*]), with more recent studies performing similar experiments in flies (*Hsu and Bhandawat, 2016*). While this approach has allowed researchers to describe the anatomy and electrophysiological responses of individual neurons, it is inherently low throughput and biased toward larger or otherwise more accessible neurons. Additionally, because experiments are typically carried out on immobile preparations, only in rare cases have investigators been able to link individual neurons to behavior (*Staudacher and Schildberger, 1998*). While recent technical and genetic advances in the model fly *Drosophila melanogaster* have improved our ability to access and manipulate individual descending neurons, to date only a handful of *Drosophila* descending neurons have been linked to specific motor outputs (*von Philipsborn et al., 2011*; *Bidaye et al., 2014*; *Zacarias et al., 2018*; *von Reyn et al., 2014*).

To assess how descending neurons control motor behaviors on a system-wide scale, it will be necessary to move beyond isolated examples and to describe the behavioral functions of large numbers of descending neurons. Our goal was to identify all the behavioral phenotypes observable in one particular setting, freely behaving flies moving within a two-dimensional arena, for many descending neurons, without any a priori expectation about the neurons' effects on behavior. *Namiki et al. (2018)* created a collection of transgenic *Drosophila* strains that target descending neurons using the split-GAL4 intersectional system (*Luan et al., 2006*; *Pfeiffer et al., 2010*) in a cell-type specific manner. We screened 130 of the sparsest lines in this collection, targeting approximately 160 neurons that are divisible into 58 distinct anatomical cell-types. Forty of these cell types consist of a single pair of bilaterally symmetric descending neurons, while the remaining 18 categories target populations of 3 to 15 descending neurons with similar neuroanatomy. We used this split-GAL4 collection to drive the expression of the red-shifted channel rhodopsin CsChrimson (*Klapoetke et al., 2014*) in specified subsets of descending neurons, allowing us to photo-activate these neurons in a temporally precise fashion. We combined these genetic reagents with a recently described method for objective, quantitative analysis of behavior (*Berman et al., 2014*) to comprehensively identify the behaviors associated with the activation of specific neurons in an unbiased fashion. Unlike supervised machine learning approaches for classifying behavior, this approach does not rely on a human-trained classifier to decide which behaviors are of interest. Instead, it captures a wide range of movements by converting high-dimensional postural dynamics into a two-dimensional map using dimensionality reduction techniques (*Berman et al., 2014*). Using this method, we associated 80% of the descending neurons in our collection with specific behaviors.

We have generated a behavioral dataset that comprehensively describes the activation phenotypes of roughly one third to one half of the total number of fly descending neurons in the context of freely walking flies. The size of this dataset has allowed us to move beyond individual examples to extract general features of descending neuron function, and therefore to consider how these neurons might encode information to modulate behaviors. We find that, with a few exceptions, descending neuron control of behavior appears to be largely modular. In addition, we find many cases in which descending neuron function is context dependent, even for a single fly confined to a two-dimensional substrate.

## Results

### Establishing a framework for large-scale analysis of descending neuron activation phenotypes

Mapping fly behavior using postural dynamics requires high temporal and spatial resolution video data from a large number of animals. Accordingly, we built a red light activation apparatus with an array of 12 USB cameras that allowed us to film 12 flies in separate chambers simultaneously at high resolution (*Figure 1A*). We crossed each split-GAL4 line to a UAS-CsChrimson line and filmed six experimental progeny that had been fed retinal, a co-factor necessary for neuronal activation via channelrhodopsin, and six genetically identical control flies whose food had not been supplemented with retinal. The flies were backlit using custom light tables, each consisting of an array of infrared and red LEDs covered by a diffuser. Each chamber was a 3 cm 'fly bubble' coated with silicone to encourage flies to remain on the flat floor of the chamber, which was in the focal plane of the camera (see Materials and methods). Each recording consisted of 30 trials of a 15 s pulse of red light followed by a 45 s recovery interval (*Figure 1A*).

If the descending neuron(s) labeled by a particular split-GAL4 line are involved in triggering, maintaining, or modulating a particular behavior, then activating these neurons with *CsChrimson* may be sufficient to activate that behavior. To identify behavioral phenotypes in an unbiased manner, we utilized the behavior mapping methods described in *Berman et al. (2014)*. First, we generated a comprehensive 'behavior space' of stereotyped actions that single flies could produce in our assay. We collected a dataset of approximately 700 million images, which included behaviors recorded from activation of descending interneuron split-GAL4 lines, previously characterized sparse GAL4 drivers (*fruitless-GAL4* and *pIP10*) that trigger courtship-related behaviors (*Stockinger et al., 2005*; *von Philipsborn et al., 2011*), and interneuron drivers targeting the flight neuropil. The additional lines that are not part of the descending neuron screen were included to sample fly behaviors as widely as possible, allowing for higher resolution mapping within the space of behaviors. We computed the behavior space by (1) aligning video images (*Figure 1A*), (2) decomposing the pixel value dynamics (which correspond to the fly's posture changes) into a low-dimensional basis set using principal component analysis (*Figure 1—figure supplement 1*), (3) projecting the original pixel values onto this basis set and transforming those values using a spectral wavelet function to produce a time series that was (4) embedded into a two-dimensional 'behavior space' (*Figure 1B*) using *t*-distributed Stochastic Neighbor Embedding (t-SNE) (*van der Maaten and Hinton, 2008*).

Each position in the behavior space corresponds to a unique set of postural dynamics. Nearby points represent similar motions, that is those involving related body parts executing similar temporal patterns. By observing the video data underlying sub-regions of the behavior space (*Figure 1—figure supplement 2* and *Videos 1–5*), we generated a human-curated version of the behavior space to aid interpretation (*Figure 1C*). In this behavior space, anterior directed movements such as eye/antennal grooming and proboscis extension are located at the top (*Video 1*). Anterior-directed foreleg movements are on the upper left side (*Video 2*). Extremely slow or still postures are on the upper right side (*Video 3*), and complex wing and abdomen movements such as body and abdomen grooming, abdomen bending and wing extension are in the center (*Video 4*). Locomotion, ranging from slow (left) to fast (right) is at the bottom (*Video 5*).

Red peaks, or density maxima, represent the fly behaviors observed most frequently in our data set. These tend to be repetitive, stereotyped behaviors, such as walking or grooming, that our analysis methodology is most sensitive at detecting. By definition, we could not detect behaviors occuring over time-scales faster than 50 Hz, the Nyquist frequency of our system. Approximately 93% of all

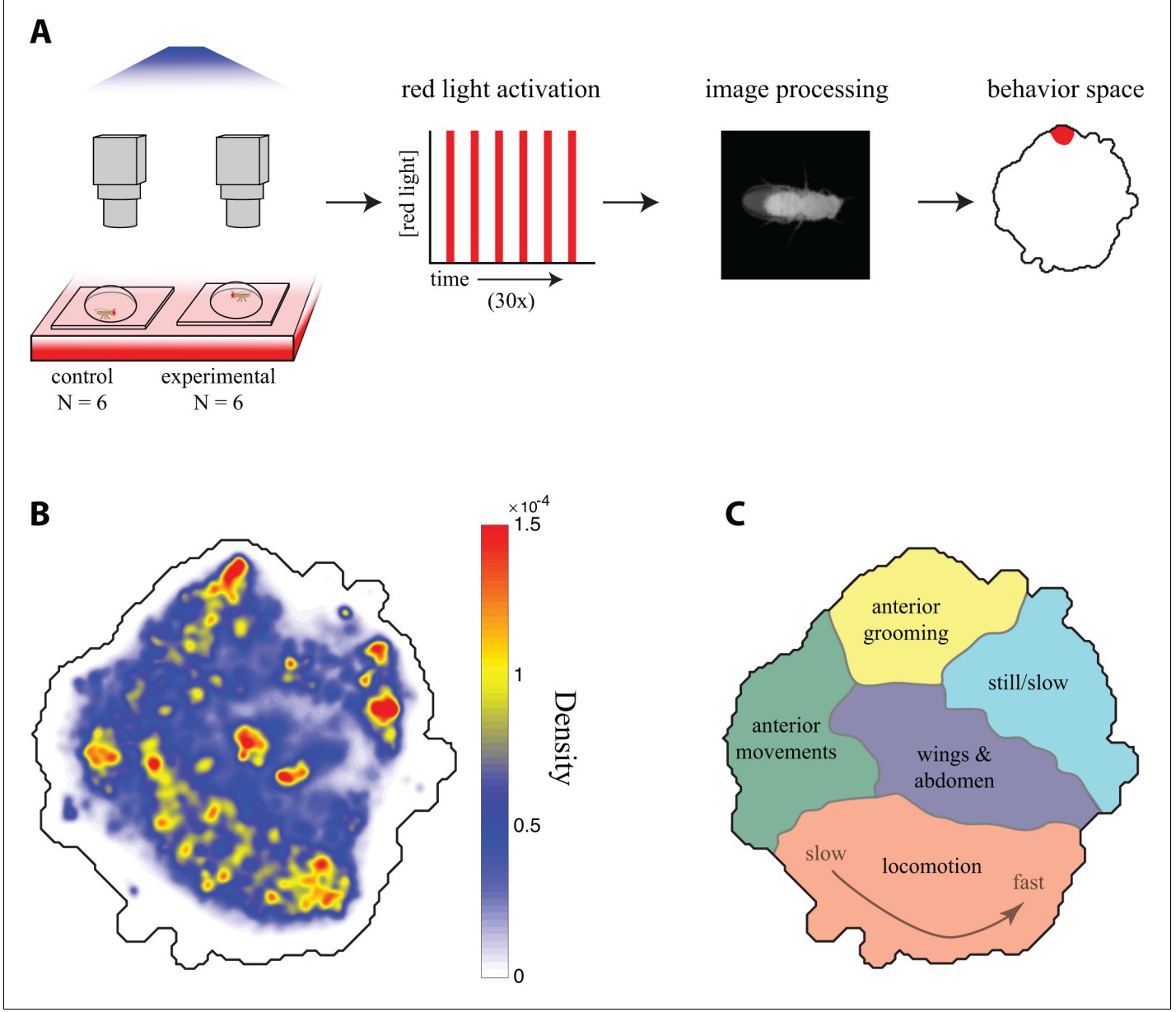

**Figure 1.** Descending neuron phenotyping pipeline and behavior space. (**A**) The red light activation rig. Six no retinal control flies and six retinal fed experimental flies were mounted in parallel in individual 3 cm diameter plexiglass bubbles on top of three custom light boards with constant 850 nm infrared light and variable 617 nm red light. The red LEDs were repeatedly turned off and on for 45 and 15 s, respectively. Each fly was filmed at 150 × 150 pixel, 100 fps resolution by a single camera. Video data was then aligned and processed, and the line was characterized for its occupancy in the descending neuron behavior space with respect to red light activation and controls. (**B**) A 2D representation of behaviors in the descending neuron video dataset was generated by applying a probability density function to all the embedded data points (scale bar), which was then convolved with a Gaussian (σ = 1.5) (**C**) Localization of various behaviors within the descending neuron behavior space seen in (**B**), based on human curation of watershedded regions in the space (*Figure 1—figure supplement 2*, *Videos 1–6*). Note that as locomotion behaviors move from left to right along the bottom of the map, the fly's running speed increases.

DOI: https://doi.org/10.7554/eLife.34275.002

The following figure supplements are available for figure 1:

**Figure supplement 1.** Postural eigenmodes used to build the descending neuron behavior space (*Figure 1B*).
DOI: https://doi.org/10.7554/eLife.34275.003

**Figure supplement 2.** Watershedded regions in the descending neuron behavior space.
DOI: https://doi.org/10.7554/eLife.34275.004

*Figure 1 continued on next page*

*Figure 1 continued*

**Figure supplement 3.** Fraction of video data points for each movie embedded in the behavior spce, during the red light stimulus window (Y axis) and during the recovery period (X axis).

DOI: https://doi.org/10.7554/eLife.34275.005

video image data points could be embedded in this space, including approximately equal fractions of frames when the red light was on or off (*Figure 1—figure supplement 3*), indicating that the majority of red light activated behaviors are well represented in the behavior space. Imaging errors, such as the fly wandering partially out of frame, are randomly distributed within the dataset.

## Entropy of behavior space density provides a quantitative and sensitive measure of optogenetic activation phenotypes

Having established a behavior space representing the full repertoire of fly behaviors that could be captured with our apparatus, we next examined which parts of this space were occupied when individual or subsets of descending neurons were optogenetically activated by CsChrimson. We focused on 130 split-GAL4 lines that targeted descending neurons with little, or no, extraneous expression in other neurons. We first considered the timing and duration of red light triggered behaviors. If descending neuron activation triggered a particular behavior represented in the behavior space, then the density of that line in the behavior space should shift into the region that represents that behavior during periods of red light activation. For example, upon red light activation, retinal-fed flies expressing *CsChrimson* in a descending neuron line targeting DNg07 and DNg08 (SS02635) groomed their heads (*Video 6*). For these lines, we identified regions in the behavior space that displayed a statistically significant shift in density for experimental flies during the first three seconds of red light compared to a window at the end of the recovery period when the red light was off (*Figure 2B,C*, *Figure 2—figure supplement 1*). This same region in the behavior space did not undergo a significant shift in the control flies (Wilcoxon rank-sum test p<0.05 using the Dunn–Šidák correction for multiple hypotheses [*Sidak, 1967*]).

Likewise, when considering densities over the whole behavior space in three second sliding windows, the experimental, but not the control, flies shifted into the head grooming region (arrowheads, *Figure 2C*).

To detect activation-elicited behaviors performed by the experimental animals and the relative timing of these dynamics compared to the red light activation, we looked for a reduction in the entropy of the behavior space density (*Figure 2A*). Entropy measures the degree of uncertainty inherent in the distribution of the flies in the behavior space. When the red light was off, flies exhibited a range of different behaviors, and the probability that they performed any one behavior was low. This resulted in a low probability density distributed throughout the behavior space and correspondingly high entropy (*Figure 2A*). Upon red light activation, the experimental fly line engaged in red light triggered behaviors at the expense of other natural behaviors. This increased the probability that a small region within the behavior space showed a relatively high density, generating a drop in entropy whose timing and duration mirrored that of the red light triggered behaviors (*Figure 2A*, *Video 7*). We can therefore use entropy as a proxy for the duration and onset of

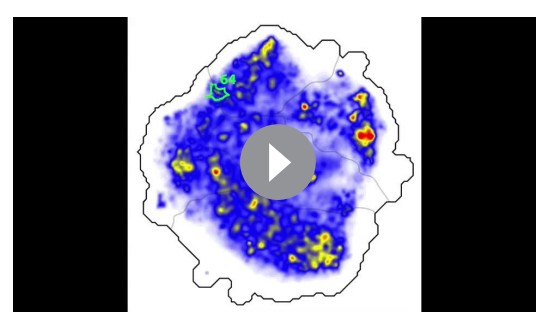

**Video 1.** Video maps of the behavioral space for anterior movements indicated in *Figure 1C*. Each movie shows a sequence of highlighted regions in the space, followed by a selection of 64 sequences of translationally and rotationally aligned fly behavior from that region. This selection is performed at random (amongst all sequences lasting at least 0.2 s within that region), showing how the raw videos are translated into the map. Because of this randomness, there are occasional image processing errors, cases where the fly goes out of the camera range, and instances of flies writhing upside-down. Note, however, that most of these sequences are sequestered to particular locations within the space, and that no descending neuron line predicts an increase in activity within these regions.

DOI: https://doi.org/10.7554/eLife.34275.006

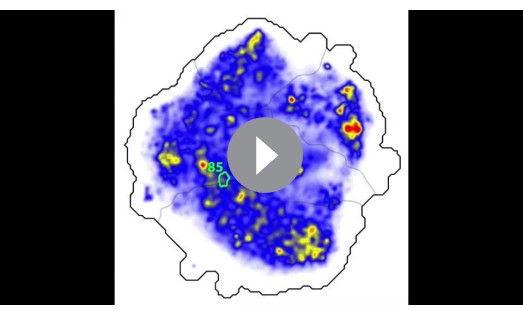

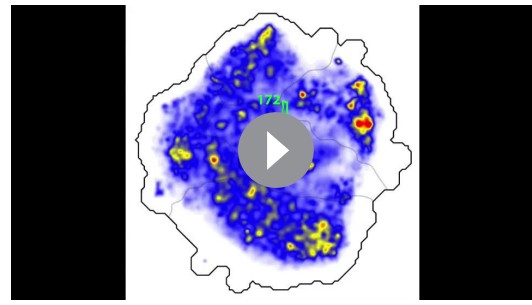

**Video 2.** Video maps of the behavioral space for anterior grooming indicated in *Figure 1C*. Each movie shows a sequence of highlighted regions in the space, followed by a selection of 64 sequences of translationally and rotationally aligned fly behavior from that region. This selection is performed completely at random (amongst all sequences lasting at least 0.2 s within that region), showing how the raw videos are translated into the map. Because of this randomness, there are occasional image processing errors, cases where the fly goes out of the camera range, and instances of flies writhing upside-down. Note, however, that most of these sequences are sequestered to particular locations within the space, and that no descending neuron line predicts an increase in activity within these regions.

DOI: https://doi.org/10.7554/eLife.34275.007

**Video 3.** Video maps of the behavioral space for still/slow behavior indicated in *Figure 1C*. Each movie shows a sequence of highlighted regions in the space, followed by a selection of 64 sequences of translationally and rotationally aligned fly behavior from that region. This selection is performed completely at random (amongst all sequences lasting at least 0.2 s within that region), showing how the raw videos are translated into the map. Because of this randomness, there are occasional image processing errors, cases where the fly goes out of the camera range, and instances of flies writhing upside-down. Note, however, that most of these sequences are sequestered to particular locations within the space, and that no descending neuron line predicts an increase in activity within these regions.

DOI: https://doi.org/10.7554/eLife.34275.008

red light triggered movements in the behavior space without needing to know, a priori, which behaviors are activated (i.e. which part of the behavior space to examine).

The region density and entropy are quantitative measurements sensitive to small changes in behavior map distribution. We therefore used these values to identify subtle phenotypes that could not be easily identified by manual inspection of the movies. For example, the activation of descending neuron DNg25 induced a short-lived rapid running phenotype (*Figure 2E and F*) that could be identified by a transient drop in entropy in the behavior space (*Figure 2D*) and a transient increase in density in the fast locomotion region of the space (*Figure 2E and F*, *Video 8*).

## Comprehensive characterization of descending neuron split-Gal4 line activation phenotypes

We searched for optogenetically induced phenotypes across the entire collection of descending neuron lines by examining the entropy time course of each line (*Figure 3*). We found that most lines displayed the largest entropy drop immediately after red light activation (*Figure 3A*). For roughly a third of the lines, this

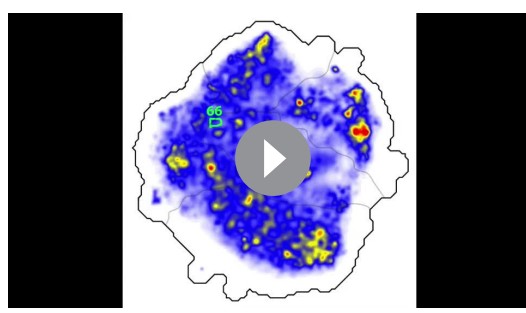

**Video 4.** Video maps of the behavioral space for wing and abdomen movements indicated in *Figure 1C*. Each movie shows a sequence of highlighted regions in the space, followed by a selection of 64 sequences of translationally and rotationally aligned fly behavior from that region. This selection is performed completely at random (amongst all sequences lasting at least 0.2 s within that region), showing how the raw videos are translated into the map. Because of this randomness, there are occasional image processing errors, cases where the fly goes out of the camera range, and instances of flies writhing upside-down. Note, however, that most of these sequences are sequestered to particular locations within the space, and that no descending neuron line predicts an increase in activity within these regions.

DOI: https://doi.org/10.7554/eLife.34275.009

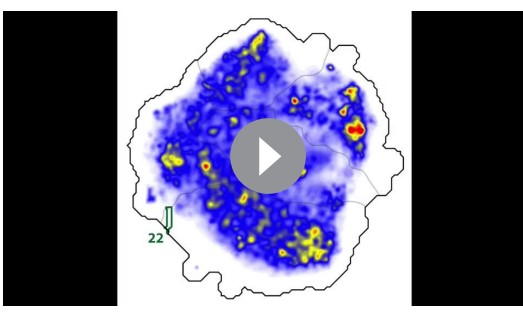

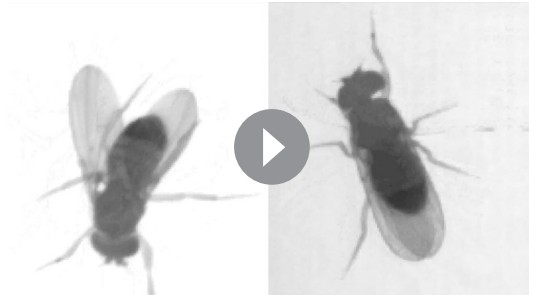

**Video 5.** Video maps of the behavioral space for locomotion indicated in *Figure 1C*. Each movie shows a sequence of highlighted regions in the space, followed by a selection of 64 sequences of translationally and rotationally aligned fly behavior from that region. This selection is performed completely at random (amongst all sequences lasting at least 0.2 s within that region), showing how the raw videos are translated into the map. Because of this randomness, there are occasional image processing errors, cases where the fly goes out of the camera range, and instances of flies writhing upside-down. Note, however, that most of these sequences are sequestered to particular locations within the space, and that no descending neuron line predicts an increase in activity within these regions.
DOI: https://doi.org/10.7554/eLife.34275.010

**Video 6.** Effect of optogenetic stimulation on descending neurons G7 and G8. This movie displays two different flies from the same line (SS02635) for 10 s before and 10 s during optogenetic stimulation . Red lights on is indicated by the text 'RED LIGHT' appearing in the lower right. Note the change in behavior immediately following stimulation.
DOI: https://doi.org/10.7554/eLife.34275.015

entropy drop persisted throughout the entire red light activation window (*Figure 3B*). For most of the rest of the lines, however, the entropy drop was transient and diminished after several seconds (*Figure 3B*). For a minority of lines, the entropy minimum occurred near the middle or end of the activation window (*Figure 3B*). These dynamics were almost fully absent in the control animals (*Figure 3C and D*).

We reviewed the raw video data for lines displaying entropy minima in the middle or end of the activation window and found that most of these flies performed some action upon red light activation, followed by a pause. This explained why the entropy was lower in the later part of the activation window, because consistent stillness generates a low entropy behavior space (see *Figure 2—source data 1* for a line by line description of phenotypes). We therefore performed our system-wide analysis using the first 3 s of the red light activation period, because this time period captured the majority of CsChrimson activated behaviors.

In our initial analysis, we looked for behaviors produced when our descending neuron lines were activated using a relatively low level of red light, (5 mW/cm$^2$). Under these conditions, 91 of the 130 lines (69%) displayed a statistically significant increase in density of some area of the behavior space. We then re-tested most of the 41 lines that did not produce a significant density increase by driving CsChrimson at higher levels by growing the flies on food containing an increased retinal concentration and exposing flies to higher intensity red light (9 mW/cm$^2$). Under these conditions, 80% of the lines that had previously displayed no phenotype produced a statistically significant increase in density in the behavior space (*Figure 2—figure supplement 2*).

Pooling the data from the low and high activation protocols, we detected statistically significant increases in the behavior space in 119 of the 130 (90%) descending neuron lines (*Figure 4A*). In 86 cases, we observed an increased density in only a single statistically significant region in the behavior space. However, some lines generated density increases in multiple non-contiguous regions of the behavior space (*Figure 4A*, examples shown in *Figure 4B–D*, *Videos 9–11*, *Figure 2—source data 1*). Many of these cases reflect multiple behaviors performed approximately simultaneously by the flies. For example, a line targeting DNp10 induced anterior reaching movements and wing flicking with similar timing (*Figure 4D*, red and blue regions, respectively).

In other cases, however, multiple activated regions reflect a stereotyped sequence of behaviors. For example, the DNp09 line shown in *Figure 4B* repeatedly ran and then paused throughout the entire 15 s activation period. The pausing observed in this case has been shown to be a defensive freezing behavior (*Zacarias et al., 2018*). The increased density in the run region of the behavior

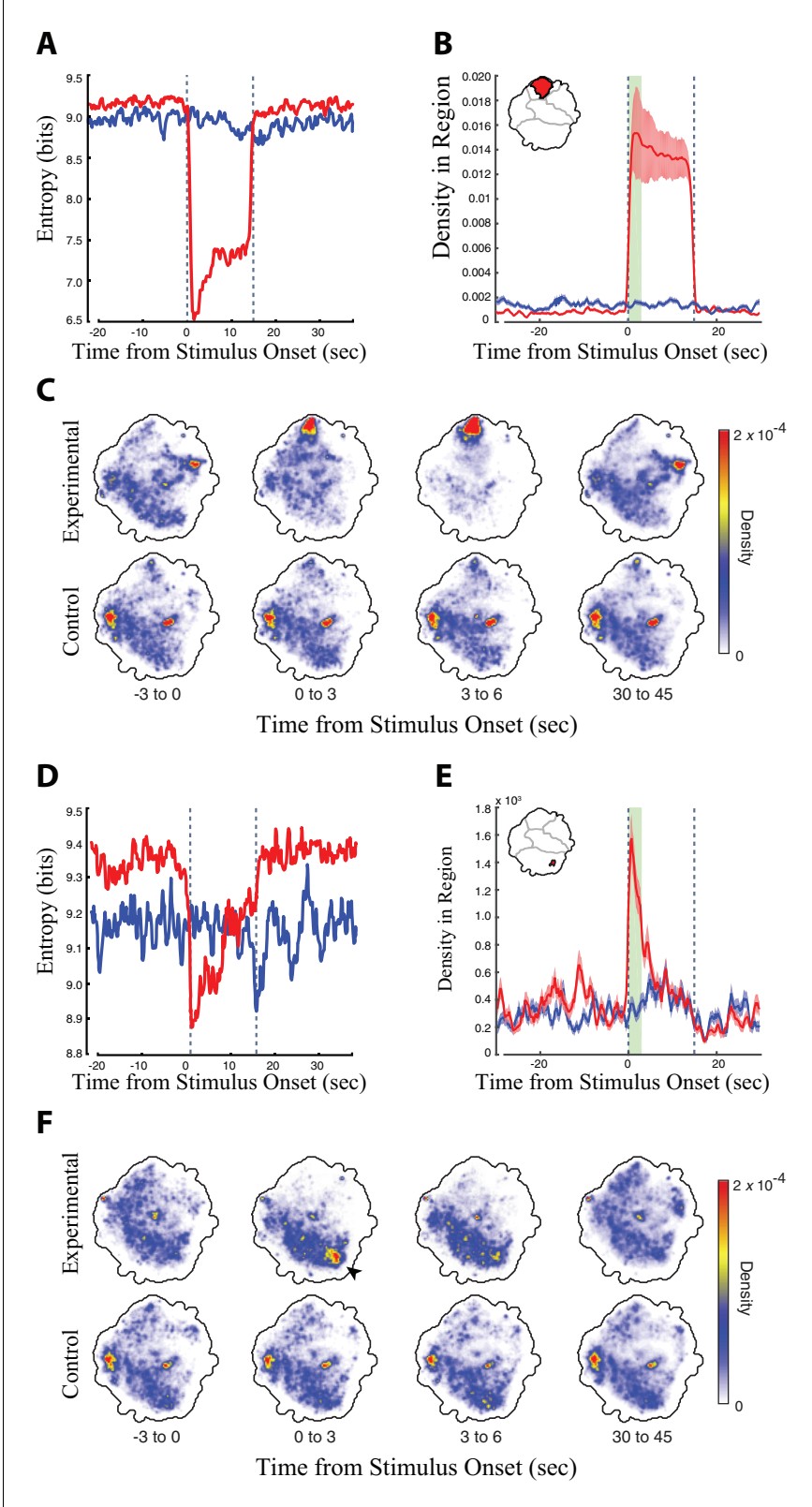

**Figure 2.** Analysis of the head grooming DNg07 and DNg08 line (SS02635) and the transient fast-running DNg25 line (SS01602). (A, D) Entropy of the distribution of the retinal fed experimental flies (red) and non-retinal fed control flies (blue) in the behavior space relative to the timing of red light stimulus onset (red light turned on at t = 0 s, the time of red light activation is indicated by dashed lines). The behavior space for experimental flies

*Figure 2 continued on next page*

*Figure 2 continued*

shows reduced entropy when the flies perform a specific set of behaviors, because flies shift from the full range of normal fly behaviors to a subset of red light-activated behaviors. (B, E) Average density ± the standard deviation in the head grooming map region indicated in red (inset, upper right) in experimental flies (red) and controls (blue) relative to red light activation. The head grooming region was calculated as the region in the map that experienced a statistically significant shift in density in experimental flies but not controls when comparing the first 3 s (green bar) of the activation period to the last 15 s of the recovery period (Wilcoxon rank sum test, $p<0.05$, using the Dunn–Šidák correction for multiple hypotheses). (C, F) Average density in the map over a series of 3 s windows (calculated from six animals, 30 trials each). Red and blue indicate regions of high and low density, respectively. The time 'before' stimulus onset is the average of 30 time periods between stimulations. We found an increase in the amount of 'Idle/Slow' dynamics for experimental flies in the interstitial times between stimulations. Thus, the differences between experimental and control animals in the time 'before' stimulation most likely reflects this increase in the amount of Idle/Slow behavior for experimental animals in the time between stimulation.

DOI: https://doi.org/10.7554/eLife.34275.011

The following source data and figure supplements are available for figure 2:

**Source data 1.** Analysis of all the descending neuron split-GAL4 lines.
DOI: https://doi.org/10.7554/eLife.34275.014
**Figure supplement 1.** Density in experimental (red) and control (blue) animals in the regions defined in *Figure 2A and D* (and B, this figure, respectively).
DOI: https://doi.org/10.7554/eLife.34275.012
**Figure supplement 2.** Choosing the red light intensity levels.
DOI: https://doi.org/10.7554/eLife.34275.013

space (*Figure 4B*, red) appeared before the increased density in the paused region (*Figure 4B*, blue), reflecting the sequential timing of the two behaviors. Running followed by freezing upon DNp09 activation is also reported by Zacarias and colleagues (*Zacarias et al., 2018*). However, the flies rapidly became asynchronous as they repeated this series of behaviors, so this behavioral series was detected as simultaneous density increases in the running and still regions throughout the red light activation window. A line targeting descending neuron DNb01 displayed a simple behavior series; flies produced an anteriorly directed twitch of the front legs when the red light was turned on (*Figure 4C*, red region), then froze for the majority of the red light activation period (*Figure 4C*, black region), and then twitched when the light was turned off (*Figure 4C*, blue region). Thus, examining the timing of density shifts illuminates the more complicated behavior series produced by red light activation. This level of analysis is provided for all lines in *Figure 2—source data 1*. To facilitate study of these activation phenotypes by others, we also provide a compilation of videos slowed down 4X showing one second before and after activation for all animals and all trials (available via Dryad https://doi.org/10.5061/dryad.fr89c0c).

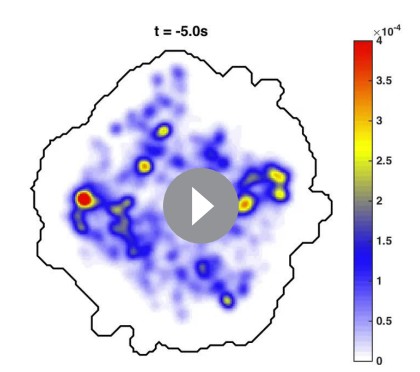

**Video 7.** Behavioral space dynamics during optogenetic stimulation for descending neurons G7 and G8 (line SS02635). Movies show the behavioral space changes from before stimulation, at the onset of stimulation, during stimulation, and after stimulation. Each movie shows the space, averaged over six experimental flies and all 30 LED cycles, for 5 s before, the 15 s during, and 5 s after the onset of stimulation (t = 0). Each frame in the video is created by finding all behavioral space positions within a 0.5 s window surrounding the displayed time and convolving each of these points with a two-dimensional gaussian of width 3 (the full movie is of height and width 210 in arbitrary units).
DOI: https://doi.org/10.7554/eLife.34275.016

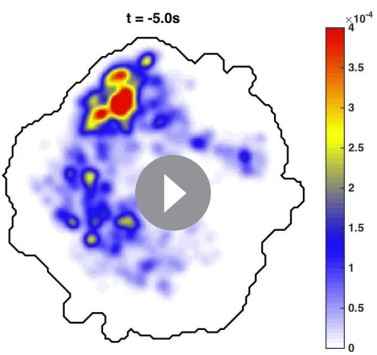

**Video 8.** Behavioral space dynamics during optogenetic stimulation for descending neuron G25 (line SS01602). These movies show the behavioral space changes from before stimulation, at the onset of stimulation, during stimulation, and after stimulation. Each movie shows the space, averaged over six experimental flies and all 30 LED cycles, for 5 s before, the 15 s during, and 5 s after the onset of stimulation (t = 0). Each frame in the video is created by finding all behavioral space positions within a 0.5 s window surrounding the displayed time and convolving each of these points with a two-dimensional gaussian of width 3 (the full movie is of height and width 210 in arbitrary units).

DOI: https://doi.org/10.7554/eLife.34275.017

## Behavioral effects of descending neuron activation are often context dependent

Why does activation of some descending neurons result in multiple, distinct behavioral outputs? One possibility is that the behavioral output of some descending neurons depends on the behavioral context of the fly when the descending neuron is activated. To address this possibility, we calculated the mutual information between the density distribution of the experimental flies in the behavior space at 1.5 to 0.5 s before the red light was turned on versus the first second after red light activation. Mutual information is a non-linear measure of the degree of dependence between two variables and is typically measured in units of bits (*Cover and Thomas, 2005*). The higher the mutual information, the more the first variable, here the behavior of flies immediately prior to red light activation as measured by their distribution in the behavior space at t = −1.5 to −0.5 s, informs the value of the second variable, the region of the behavior space occupied in the first second of red light activation.

We found that all experimental animals displayed non-zero mutual information between the pre- and post-stimulation behaviors (*Figure 5A*). In addition, for most lines, more information was available in the experimental flies than in the controls (*Figure 5B*). This means that even in those cases where red light activation produced only one significant region in the behavior space, the fly's activity prior to red light activation influenced whether or not it performed the behavior. However, lines with multiple red light activated regions in the behavior space were also those with a relatively high level of mutual information (*Figure 5A–B*). Thus, a given fly's behavior before red light activation was highly informative of which behavior that fly would perform after red light activation, as indicated by the different significantly activated regions in the behavior space. *Figure 5C* displays this phenomenon for one of the lines with the highest mutual information, SS02542 (asterisk in *Figure 5A and B*, also shown in *Figure 4C*). Here, if the flies were performing an action in the wing/abdomen movement regions of the behavior space prior to stimulation, then they were likely to perform an anterior movement (region 1) immediately following stimulation. Similarly, flies performing anterior grooming were likely to transition to the small anterior twitch region (region 2), and flies that were initially still tended to remain still post-stimulation (region 3).

## Individual descending neurons produce mainly stereotyped, modular behaviors

So far, we have analyzed split-GAL4 lines as if they were a proxy for individual descending neurons or anatomical classes of descending neurons. However, these lines vary in both their strength of expression and in the number and identity of additional cells labeled. To estimate phenotypes for individual descending neurons, we averaged the behavior space densities of multiple lines for those cases where we had multiple lines targeting the same descending neuron (*Figure 6*). Using this method, and combining it with those descending neurons for which we had only a single representative split-GAL4 line, we estimated phenotypes for 47 of the 58 descending neuron cell types. We have also included six lines and line averages that target two different types of descending neurons cleanly, but for which we have no lines that target each type individually. Twenty-six descending

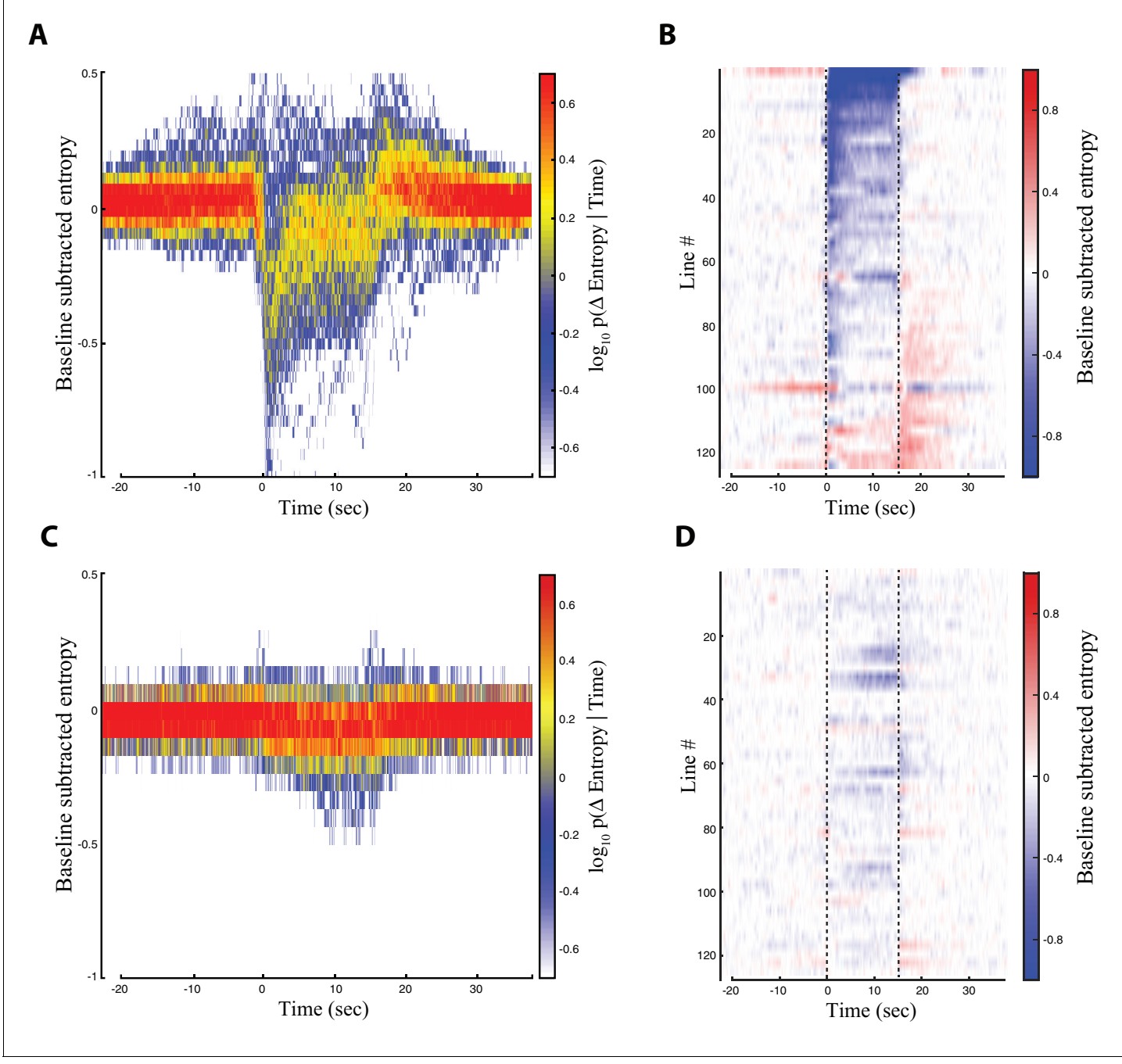

**Figure 3.** Entropy for all the descending neuron split-GAL4 lines. Trials were aligned such that the onset of red light activation occured at t = 0 s. The red light was turned off at t = 15 s. (A) Baseline subtracted entropy (Y-axis) versus time (X-axis) for all experimental animals and all trials. Colors indicate a probability distribution describing the entropy of experimental animals at a given time in the trial. (B) Entropy levels of experimental animals over the course of the aligned trials (X-axis) shown by line (Y-axis). Warm and blue colors indicate high and low entropy, respectively. (C) and (D) Same as (A) and (B), but for the control animals. Lines are presented in the same order as in (B) and (D).

DOI: https://doi.org/10.7554/eLife.34275.018

neurons drove locomotion phenotypes and ten drove anterior directed foreleg movements. We also identified six new descending neurons that triggered wing and abdomen movements (plus the previously published pIP10 [*von Philipsborn et al., 2011*]), two that drove anterior grooming, one that drove abdomen stroking, and four that drove still or slow behaviors.

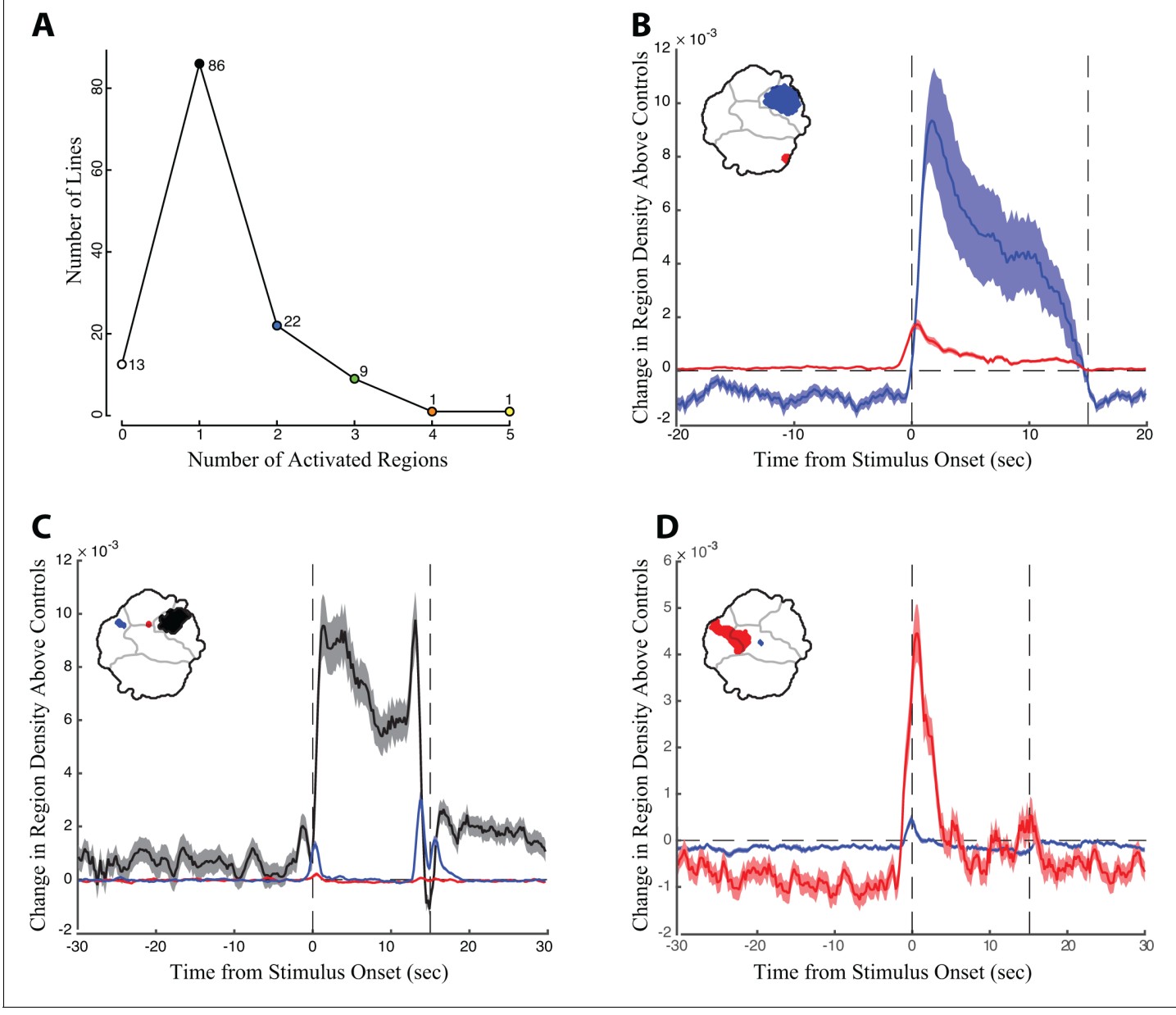

**Figure 4.** The timing of density shifts in descending neuron lines that occupy multiple behavior space regions. (**A**) Most lines (86) increase density in only one region of the behavior space upon red light activation. However, some lines occupy multiple regions in the behavior space. (**B–D**) Examples of lines that occupy multiple discontinuous regions upon red light activation. Time is indicated on the X-axis, with the red light turned on at T = 0 s and off at 15 s. Change in density in the color coded regions in the experimental animals above the controls is indicated on the Y-axis. (**B**) Line SS01540 targeting descending neuron DNp09; (**C**) SS02542 targeting descending neuron DNb01; (**D**) SS01049 targeting descending neuron DNp10. In many cases, as shown here, we observed an increase in region density that appears to occur immediately before the stimulus onset. This increase in density probably results from temporal smoothing of approximately 1 s that is introduced through the wavelet transformation of the behavioral analysis and not due to behavioral anticipation of the light stimulus.

DOI: https://doi.org/10.7554/eLife.34275.019

In general, we found that activation of each type of descending neuron drove behaviors that mapped to a relatively small region of the behavior space. For example, some descending neurons drove slow locomotion, whereas others drove fast locomotion. Only a few, such as DNa01, DNa02 and DNp26, seemed to produce a global increase in locomotor activity. Likewise, we found descending neurons that produced different types of grooming, such as head grooming (DNg07

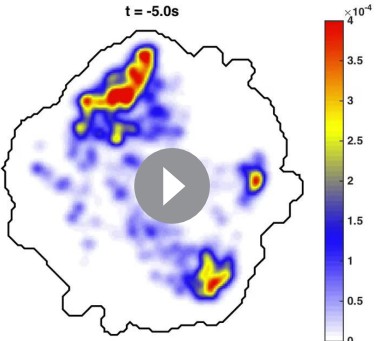

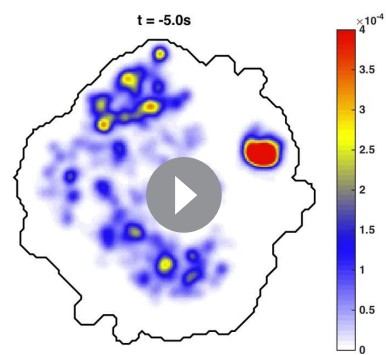

**Video 9.** Behavioral space dynamics during optogenetic stimulation for descending neuron G9 (line SS01540). These movies show the behavioral space changes from before stimulation, at the onset of stimulation, during stimulation, and after stimulation. Each movie shows the space, averaged over six experimental flies and all 30 LED cycles, for 5 s before, the 15 s during, and 5 s after the onset of stimulation (t = 0). Each frame in the video is created by finding all behavioral space positions within a 0.5 s window surrounding the displayed time and convolving each of these points with a two-dimensional gaussian of width 3 (the full movie is of height and width 210 in arbitrary units).
DOI: https://doi.org/10.7554/eLife.34275.020

**Video 10.** Behavioral space dynamics during optogenetic stimulation for descending neuron B1 (line SS02542). These movies show the behavioral space changes from before stimulation, at the onset of stimulation, during stimulation, and after stimulation. Each movie shows the space, averaged over six experimental flies and all 30 LED cycles, for 5 s before, the 15 s during, and 5 s after the onset of stimulation (t = 0). Each frame in the video is created by finding all behavioral space positions within a 0.5 s window surrounding the displayed time and convolving each of these points with a two-dimensional gaussian of width 3 (the full movie is of height and width 210 in arbitrary units).
DOI: https://doi.org/10.7554/eLife.34275.021

and DNg08, and DNg12) or abdomen grooming (DNp29), different types of anterior reaching movements (DNg10 versus DNg13) and different types of slow movements (e.g. DNd02 versus DNp02).

## Discussion

Using optogenetic activation and automated behavioral quantification, we assigned behavioral phenotypes to 80% of the descending neurons cell types in our collection of lines, or one third to one half of the estimated total number of descending neurons present in the fly. Using a dataset of this scope, it is possible for the first time to move beyond isolated examples to consider systems-level trends in how descending neurons control behaviors. We found that activation of most descending neurons drove stereotyped behaviors, that in many cases multiple descending neurons activated the same behaviors, and that activated behaviors were often dependent on prior behavior states.

There are several, non-mutually exclusive ways a limited number of seemingly highly modular descending neurons could encode the wide range of behaviors undertaken by freely moving animals. First, descending neurons could be more important for triggering and maintaining behaviors than for controlling individual details of a given motor program (*Heinrich, 2002*). Many motor programs, particularly those controlling repetitive, rhythmic actions such as walking or stridulation, can function in the absence of descending control ([*Bentley, 1977*; *Kien, 1983*], for a review on walking circuits see [*Ritzmann and Bü Schges 2007*]). For example, *Hedwig, 1992* identified two pairs of descending neurons that control stridulation in grasshoppers. In this system, tonic activation of the descending neurons was sufficient to induce and modulate the activity of the stridulation central pattern generator in the thorax, indicating that the descending neurons play only a limited role in patterning stridulation. Several of our lines, including the DNg07 and DNg08 head grooming line (*Figure 2A and B*), appear to reflect a similar phenomenon, driving a repeated stereotyped behavior during the entire CsChrimson activation window.

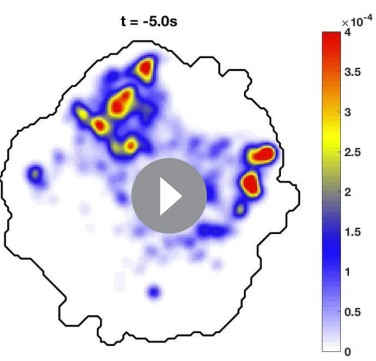

t = -5.0s

**Video 11.** Behavioral space dynamics during optogenetic stimulation for descending neuron P10 (line SS01049). These movies show the behavioral space changes from before stimulation, at the onset of stimulation, during stimulation, and after stimulation. Each movie shows the space, averaged over six experimental flies and all 30 LED cycles, for 5 s before, the 15 s during, and 5 s after the onset of stimulation (t = 0). Each frame in the video is created by finding all behavioral space positions within a 0.5 s window surrounding the displayed time and convolving each of these points with a two-dimensional gaussian of width 3 (the full movie is of height and width 210 in arbitrary units).

DOI: https://doi.org/10.7554/eLife.34275.022

Second, behaviors might be controlled not by single descending neurons acting as command neurons, but by combinations of descending neurons acting in concert (*Heinrich, 2002*). For example, several previous studies have illustrated that multiple descending neurons control the same specific behaviors (*Kien, 1983*; *Griss and Rowell, 1986*; *Rowell and Reichert, 1986*; *Gronenberg and Strausfeld, 1990*; *Kien, 1990*; *Milde and Strausfeld, 1990*; *Hensler, 1992*; *Kanzaki et al., 1994*; *Staudacher, 2001*). In these cases, it has been suggested that sufficiently strong stimulation of one neuron in a command cohort or module is sufficient to recruit the activity of the other descending neurons, resulting ultimately in triggering of the behavior (*Larimer, 1988*). Evidence from neuroanatomy further supports this hypothesis. For example, roughly a third of described descending neurons appear to have unique projection patterns in *Drosophila* and *Calliphora*, but the rest share common input and/or output regions in the brain and ventral nerve cord with other descending neurons, suggesting they may act in concert (*Gronenberg and Strausfeld, 1990*; *Milde and Strausfeld, 1990*; *Namiki et al., 2018*). Our data do not allow us to definitively address this question. However, the large number of descending neurons that drive similar patterns of fast locomotion, slow locomotion and anterior reaching in our dataset suggests that, for these motor circuits at least, these neurons may act in concert. Alternatively, it is possible that many of these descending neurons modulate distinct aspects of these motor programs, a potential dynamic that we did not address here.

Third, another way to generate behavioral complexity is by coding different behaviors via combinations of descending neurons. We examined a few lines that target multiple descending neurons. We compared behaviors produced by these 'multi-hit' split-GAL4 lines with lines that targeted the individual neurons and found only weak evidence for the emergence of new behaviors when descending neurons were triggered in combination. For example, both DNa05 and DNd02 produce slightly different phenotypes when activated in combination with DNa07 and DNd03, respectively, as compared to when lines targeting these neurons are activated alone (see *Figure 2—source data 1*). However, our collection contains, by design, few lines driving expression in combinations of descending neuron types. Therefore, further exploration of this idea will require the generation and characterization of additional lines.

Finally, descending neurons could be re-used in multiple behavioral contexts. While there are to date no published examples of a single descending neuron triggering different context-dependent behaviors, there are multiple cases in which descending neurons exhibit different physiological responses depending on the state of the animal (e.g. walking, flying, courting, etc.) (*Olberg, 1983*; *Strausfeld and Bassemir, 1985*; *Olberg and Willis, 1990*; *Böhm and Schildberger, 1992*; *Staudacher and Schildberger, 1998*; *Hedwig, 2000*; *Staudacher, 2001*; *Zorović and Hedwig, 2011*). Our results strongly support a role for context dependency for two reasons. First, the high level of mutual information between behaviors immediately before and after red light activation seen in lines that have multiple red light activated regions indicates that even within the relatively simple confines of our assay, the behavior of the fly immediately before descending neuron activation biases the behavioral output in many cases. Second, our observation that substrate specific behaviors, such as foreleg tapping, reaching, and locomotion are strongly represented in our dataset, while flight and courtship behaviors are less prevalent suggests that descending neuron outputs

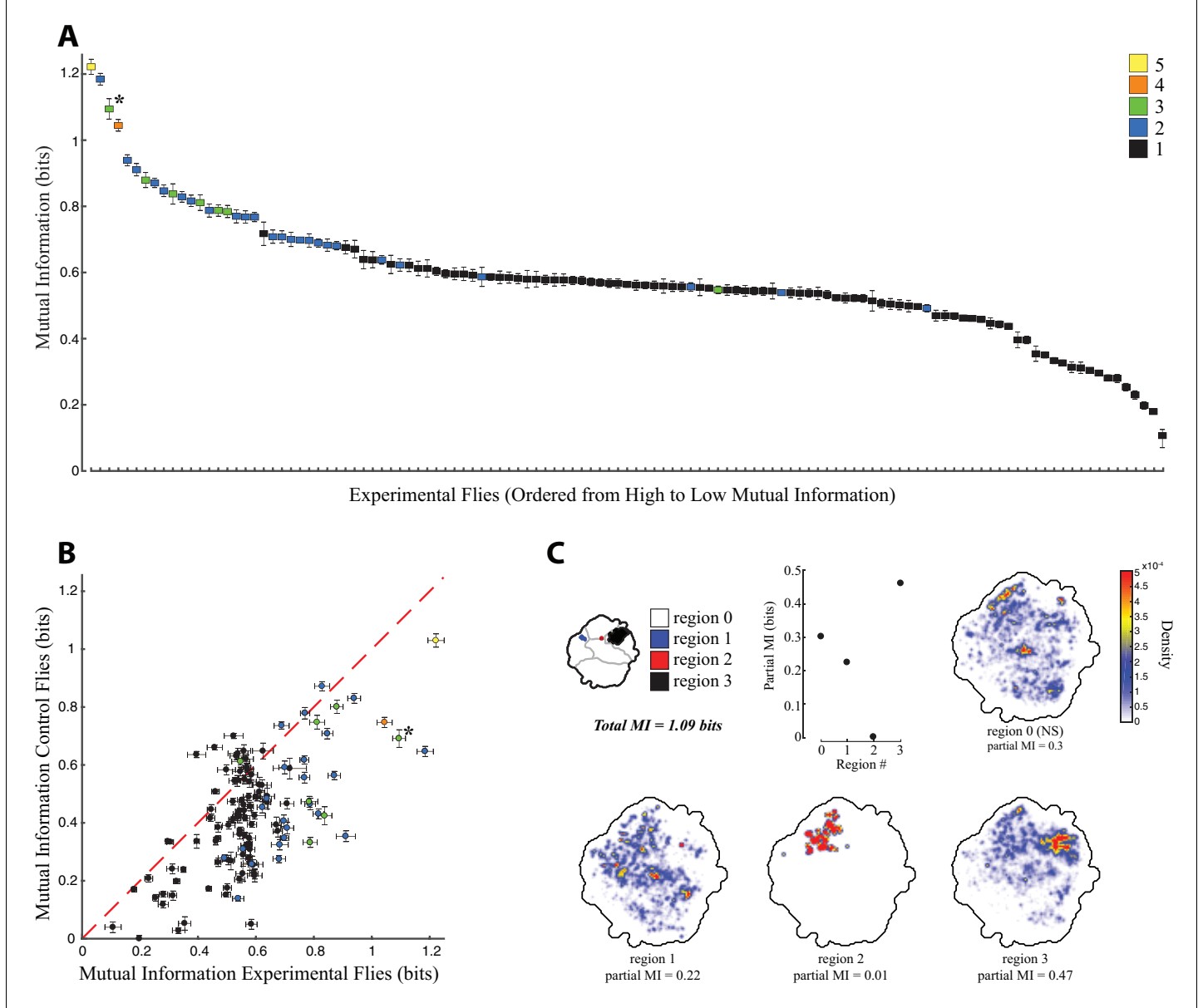

**Figure 5.** Mutual information between behaviors performed before and during red-light activation. (**A**) Mutual information between pre-stimulation and post-stimulation behavior for experimental flies, calculated using the density of the flies in the behavior space at t = −1.5 to −0.5 s prior to red light activation versus density at t = 0 to 1.0 s after red light activation. Y-axis indicates the mutual information, X-axis is all the lines in order of most to least mutual information, error bars indicate the standard deviation. Lines are color coded by the number of significant regions produced by red light activation, the key is indicated in the upper right, line SS02542 is indicated by an asterisk. (**B**) Mutual information between pre-stimulation and post-stimulation behavior of experimental flies (X-axis) plotted against the same quantity for the non-retinal fed control flies (Y-axis), with the same color coding as (**A**). Most experimental animals display stronger mutual information than control flies, implying that descending neuron activation makes fly behavior more dependent on prior state than in control flies. (**C**) Partial mutual information for SS02542, showing the density at t = −1.5 to 0.5 that has mutual information with the different regions of the behavior space after red light activation indicated in the key in the upper left portion of the figure.
DOI: https://doi.org/10.7554/eLife.34275.023

may be context dependent. By forcing the flies to remain on a two-dimensional substrate in isolation, we may have observed predominantly indirect results of behaviors that would normally take place in a different context. For example, when we activated a line expressing in descending neuron DNp01, the giant fiber, a neuron known to elicit a rapid escape response initiated by a jump when optogenetically activated (*Lima and Miesenböck, 2005*) we only detected the flies running after

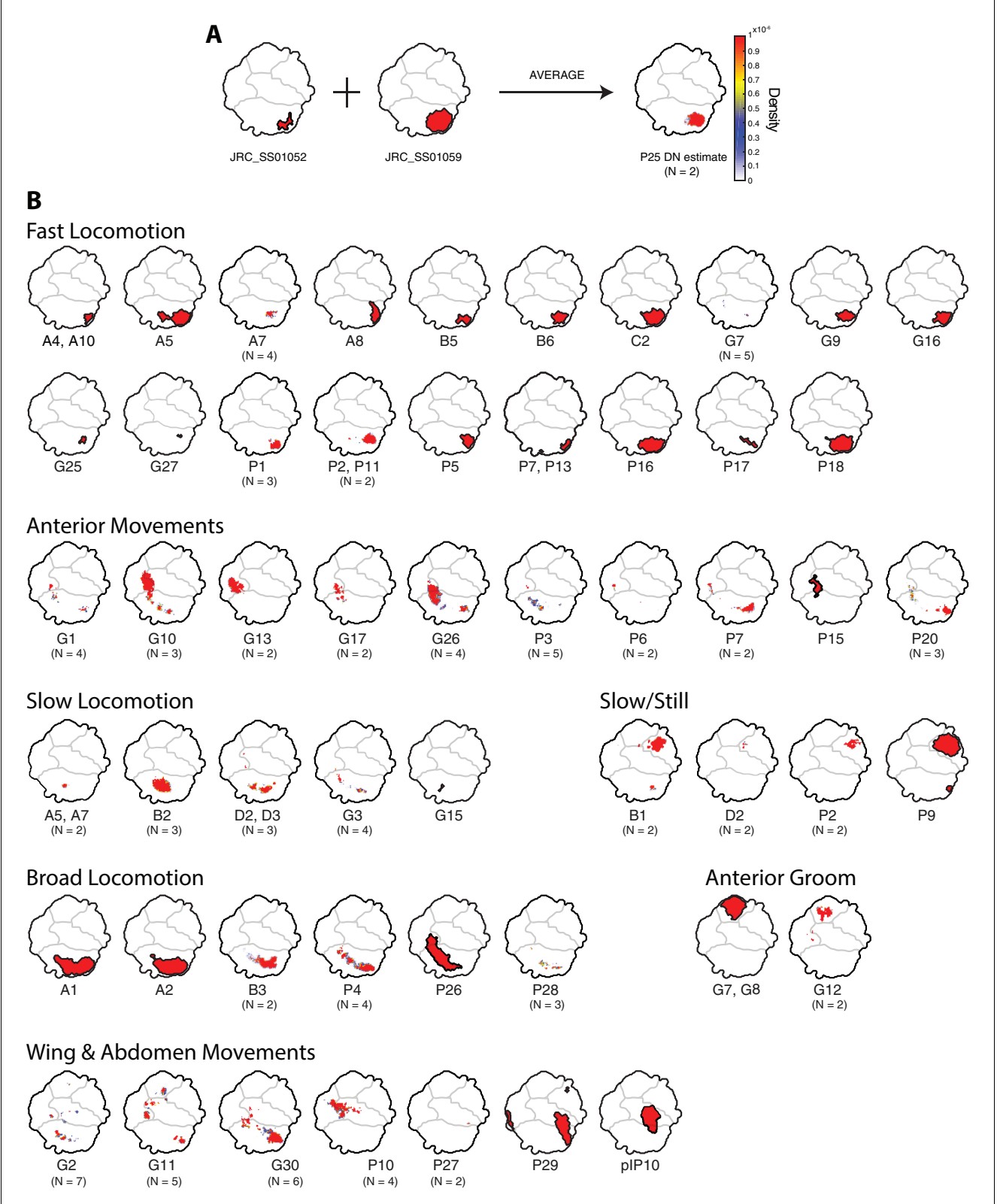

**Figure 6.** Averages of representative lines for individual descending neurons. Descending neurons are organized according to the region of the behavior space that they activate (*Figure 1*). (**A**) Example illustrating the averaging of two lines to produce an estimated phenotype. Colors indicate the degree to which particular regions are represented on average. Red regions are highly represented, blue less so, and white not at all. (**B**) Phenotypes

*Figure 6 continued on next page*

*Figure 6 continued*

for 53 of the 58 descending neurons in the collection, plus pIP10 (*von Philipsborn et al., 2011*). Some descending neurons are represented by a single clean line, others by averaging multiple clean lines. For averages, the number of lines are indicated under the descending neuron name.

DOI: https://doi.org/10.7554/eLife.34275.024

returning back to the ground because the jump was too fast (~30 ms) to be detected in our assay. It is also possible that some of the descending neurons we screened are never naturally activated in the two-dimensional context of walking and that proprioreceptive feedback may have generated abnormal behaviors in our assay.

There are futher limitations of our dataset. First, behaviors performed more quickly than the Nyquist frequency of 50 Hz for our movies could not be detected. Second, we assayed only behaviors that can be activated when flies are standing and walking. Descending neurons controlling flight-related behaviors, for example, would not be detected. Third, we assayed only males, so any female-specific behaviors may not be identified. Finally, we assayed solitary flies, so any behaviors dependent on social interactions, for example courtship, may not have been detected.

Our objective, quantitative assessment of a descending neuron activation screen provides a foundation for understanding descending neuron functions more broadly. Using similar analytical approaches to study the results of descending neuron activation and inactivation in other behavioral settings in the future will broaden our understanding of how descending neurons direct motor patterns in specific behavioral contexts and reveal how the fly's rich behavioral repertoire can be encoded with only a few hundred descending neurons.

## Materials and methods

### Fly stocks and fly handling

The descending neuron split-GAL4 driver collection is described in *Namiki et al. (2018)*. Male flies were crossed to virgin females carrying *20xUAS-CsChrimson-mVenus* (*Klapoetke et al., 2014*) integrated into the *attP18* landing site (*Markstein et al., 2008*) and transferred to Dickson lab power food (1L water, 10 g agar, 80 g Brewer's yeast, 20 g yeast extract, 20 g peptone, 30 g sucrose, 60 g dextrose, 0.5 g $MgSO_4{*}6H20$, 0.5 g $CaCl_2{*}2H20$, 6 mL propionic acid and 7 mL 15% Nipagin). For the initial screen, experimental animals were raised on power food supplemented with 0.2 mM retinal. This concentration was increased to 0.4 mM for animals that were re-assayed at a higher light intensity. All flies (except parental stocks) were handled under 453 nm blue LEDs and reared in dark blue acrylic boxes (acrylic available from McMaster-Carr, # 8505K84) at 22°C on a 12 hr lights on:12 hr lights off day:night cycle. Individual male flies were collected upon eclosion and housed singly in 2 mL wells in a 96-well 'condo,' with power food (with or without retinal) deposited in the bottom of each well, which was sealed at the top with an airpore sheet (Qiagen #195761). Flies were imaged at age 7–12 days, within 4 hr of lights on.

### Data collection

Single flies were loaded into individual trays made from 4.5 mm clear acrylic topped with a fly 'bubble' 3 cm in diameter and 4 mm at its tallest point, which was vacuum molded from clear 0.020' PETG thermoform plastic (WidgetWorks Unlimited) (*Berman et al., 2014*; *Klibaite et al., 2017*). PETG was placed in a frame, heated in a Oster Convection Bake pizza oven set at 350 °F until the plastic started to deform (about 20 s), then placed on a vacuum former (WidgetWorks Unlimited). To further encourage the flies to remain on the two-dimensional acrylic surface, the bubbles were coated with Sigmacote siliconizing reagent 1 day prior to imaging and lightly wiped with ethanol to remove the excess silicone.

For each descending neuron split-GAL4 line, six retinal-fed experimental animals and six non-retinal-fed control animals were imaged simultaneously. For imaging, flies were placed in individual fly bubbles atop custom light tables (three identical light tables, each imaging four flies). These tables consisted of a custom light board topped with a 0.75' 3D printed white plastic standoff that was lined with infrared reflective tape, and which was capped with a diffuser made from 0.125' white plexiglass acrylic (available from eplastics.com, # ACRY24470.125PM24 × 48), which had 50% light

transmittance. The light board itself consisted of an array of 256 IR LEDs (Osram Opto SFH 4050-Z, 850 nm wavelength) arrayed in a 16 × 16 pattern, spaced 7.14 mm apart, and 64 red LEDs (Philips Lumileds, LXM2-PD01-0050, 627 nm) arranged in an 8 × 8 pattern, spaced 14.28 mm apart. IR and red LED intensity was controlled separately by 0–2.5V control voltages, yielding 0-100mA for the IR LEDS and 0–400 mA for the red LEDs. We set the IR LEDs to 1V, which provided even illumination without overheating the flies. We used 0.2V (4.5 mW/cm$^2$) red light in the initial screen and 1.0V (9 mW/cm$^2$) red light when re-screening a subset of lines. All three light tables were connected to a 68-Pin unshielded I/O connector block (National Instruments, CB-68LP), then to an M Series multifunction DAQ board (National Instruments, NI USB-6281), so that all tables could be run simultaneously from a single computer. Each light table was topped with a 10″ square frame constructed from off the shelf parts from Thorlabs, which supported four 1.3 MP grayscale USB cameras (Point Grey FL3-U3-13Y3M-C, one camera per fly) on optical rails, whose X/Y/Z coordinates could be adjusted relative to the fly bubble. Each camera was fitted with an HR F2/35 mm lens from Thorlabs and an 800 nm longpass filter (Thorlabs, FEL0800). Each set of four cameras was connected to a separate Dell Precision T3600 Tower Workstation containing two 100 GB internal solid state drives, so that two cameras wrote to each solid state drive.

Cameras were programmed using NI-MAX and custom software written in Labview (National Instruments). Data acquisition and the LED light tables were controlled by custom software written in Labview. In brief, a master computer ran a single program that (1) turned the red and infrared (*Berman et al., 2014*) LEDs on for all three light tables, the former running a program of 15 s, then off for 45 s, for 30 cycles; (2) started all 12 cameras recording; (3) recorded the position of the fly's centroid for each frame for each camera; and (4) grabbed the frame number for each camera over the network every 2–3 frames, and wrote the frame number and red LED status from the light tables to a single text file. All movies were recorded as uncompressed avi files at 100 frames per second. Each camera was set to 1024 × 1024 pixel resolution that encompassed the entire 3 cm arena. However, flies were tracked using a blob detector and only a 150 × 150 pixel box centered on each fly was saved and used for analysis.

To facilitate direct examination of individual activated behaviors, we have provided a subset of the complete dataset, consisting of 1 s before and after stimulation for all flies and all trials, which is deposited at Dryad (doi:10.5061/dryad.fr89c0c).

## Behavior space generation

Our approach for generating a behavior space largely follows the methodology originally described in *Berman et al. (2014)*, which describes much of the procedure in additional detail. We first segmented flies using Canny's method for edge detection (*Canny, 1986*) and morphological dilation to find the outline of the fly. All pixels within the corresponding closed curve were considered part of the fly. We assumed that all flies had identical morphology but variable sizes. We calculated a rescaling factor for each fly by segmenting 100 randomly selected images from a single fly and finding the pixels belonging to that fly's body (head, thorax, and abdomen) in each of them, ignoring pixels associated with the wings and legs. Body pixels were assigned via a two-component Gaussian mixture model, and the average value of the number of pixels was chosen as the body area. All frames from a single movie were then uniformly re-scaled to make the number of body pixels in the average image equal to that in a reference image of a fly. We then rotationally aligned segmented, recalled images by finding the maximal angular cross-correlation of the magnitudes of the two-dimensional polar Fourier transforms between the image and a reference image. This reference image was common to all aligned images. Translational registration was then performed by maximizing the spatial cross-correlation.

Postural decomposition was performed as described in *Berman et al. (2014)*. Images were Radon-transformed using a two-degree spacing, and the 9781 Radon-space pixels that contained the most variance were kept for further analysis (>95% of the total variance). We then performed principal components analysis (PCA) on these data, keeping the 50 modes capturing the most variance (>90% of the total variance). We projected the segmented and aligned images onto the found eigenvectors to create a set of time series that were representative of the postural movements of the fly. To obtain dynamic information about these time series, we applied a Morlet continuous wavelet transform to these time series. We transformed each mode separately, using 25 frequency

channels that were dyadically spaced between 1and 50 Hz, retaining only the amplitudes of the resulting complex numbers.

Low-dimensional embedding of these wavelet time series using t-Distributed Stochastic Neighbor Embedding (t-SNE) (*van der Maaten and Hinton, 2008*) largely followed the approach in *Berman et al. (2014)* as well. A distance metric between points in time was calculated via the the Kullback-Leibler divergence (*Cover and Thomas, 2005*) between their associated normalized mode-frequency spectra. Because this data set contains several orders of magnitude more data than can be calculated through brute-force minimization of the t-SNE cost function, we used the sub-sampling technique described in *Berman et al. (2014)* to identify 600 representative data points from each of the recording sessions. From here, points were randomly assigned subsequent groupings such that each of these groups contained 36,000 data points. The same sub-sampling process was performed amongst these data points, but now keeping twice as many data points as in the previous iteration. This process was repeated until a data set of 36,000 points was obtained. We minimized t-SNE for this data set to create a low-dimensional embedding. We used the re-embedding procedure described in *Berman et al. (2014)* to include data from outside the 36,000-point training set into the embedding, resulting in the overall density seen in *Figure 1*.

## Statistical analysis

Our main goal for the statistical analysis of the behavior space data was to isolate regions of the map that were significantly affected by optogenetic stimulation. Here, we assessed significance by (1) comparing the flies' behaviors when the LED was on versus when the LED was off, and (2) requiring that the effect of the LED stimulation be larger in the experimental flies than in the control flies. Specifically, we compared the flies' behaviors during the first three seconds of stimulation (t=0s to t=3s, where the LED turns on at t=0) to their behavior between stimulation (t=30s to t=45s). To statistically assess whether a particular region of the behavior space was significantly affected by the stimulus, we first defined $\rho_{i,n}^{on}(x,y)$ to be the average behavior space density for fly $i$ during the $n$th cycle at location $(x,y)$ during the first 3 seconds of excitation and $\rho_{i,n}^{off}(x,y)$ to be the same, but during the 15s window furthest from the stimulation. We then tested whether $\rho_{i,n}^{on}(x,y)$ was significantly different from $\rho_{i,n}^{off}(x,y)$ through a Wilcoxon rank sum test with Šidák corrections (p<0.05 after corrections) (Šidák 1967). To calculate the number of corrections, we conservatively assumed that the number of measurements was equal to $2^H$, where $H$ was the entropy of the mean density of the behavior space. This is likely an over-estimate of the number of comparisons, but it provides an upper-bound for the number of distinctions that could be made.

To compare the effect of the optogenetic stimulus on the experimental flies to that of the effect on the control flies, we computed the quantity $\chi_{i,n}(x,y) = \rho_{i,n}^{on}(x,y) - \frac{1}{2}\left(\rho_{i,n-1}^{off}(x,y) + \rho_{i,n}^{off}(x,y)\right)$, which was the behavior space density during light stimulation compared to the average of the two preceding time periods with no light stimulation. We thus assessed statistical significance by using a Wilcoxon rank sum test with Šidák corrections (p<0.05 after corrections) to compare $\left\{\chi_{i,n}(x,y)\right\}_{i \in experimental\ flies}$ with $\left\{\chi_{i,n}(x,y)\right\}_{i \in control\ flies}$. For a point, $(x,y)$, in the behavior space to be considered significantly affected by the stimulus, we required that both of these tests—within experimental flies test and experimental versus control flies test—yielded a significant result.

Behavior activation maps for individual descending interneurons (*Figure 6*) were calculated by averaging together the maps of significant activations ($E\left[\chi_{i,n}(x,y)\right]_{i,n} > 0$) from each of the lines exciting that neuron.

Stimulation-response entropy curves (*Figure 2A and D*) were generated by first aligning each time point to its associated phase within the 60 second LED on-off cycle. For each phase within the cycle, we found all embedding points from all relevant trials that were detected within $\pm 200$ ms (using periodic boundary conditions). We then generated a histogram of these points, normalized and convolved the resulting values with a symmetric two-dimensional Gaussian of width $\sigma = 2$, to generate a probability density function, $p_t(x,y)$. From this, the entropy curve value at phase $t$ was given by $H(t) = \int dx\, dy\, p_t(x,y)\log p_t(x,y)$. We then pooled data from all individuals of a specific type together (i.e. all control flies from a given line or all experimental flies from a given line) to calculate these curves.

Mutual information between pre-stimulus behavior space densities and post-stimulus regions was computed by numerically integrating the integral:

$$MI\left(\rho_{pre};R_{post}\right) = \sum_{k=0}^{m} \int d\vec{x}\, p_{pre}\left(\vec{x}|R_k\right) log_2 \frac{p_{pre}\left(\vec{x}|R_k\right)}{\sum_{l=0}^{m} p_{pre}\left(\vec{x}|R_k\right)p(R_k)},$$

where $p_{pre}\left(\vec{x}|R_k\right)$ is the conditional probability of observing the fly's behavior to be at location $\vec{x}$ between 1.5 and 0.5 seconds before the stimulus onset and $p(R_k)$ is the probability that the fly transitions to region $R_k$ following the stimulus onset. Finite data-size corrections were performed by drawing subsets of the data with replacement and extrapolating to an infinite number of trials, and error bars were generated by extrapolating the calculated variance in a similar manner (*Bialek, 2012*). The region of transition for each trial was assigned by finding the mode of the behavior space distribution during the first second subsequent to the onset of the stimulus. If the location of the mode of the distribution for that trial was within or closer than 5-pixels to the edge of a region, it was assigned to that region, unless another region was closer. Trials not assigned to any of the regions were given a 'zero' label, as reflected in the previous equation.

To provide a sense of scale, if there are $N$ significantly activated regions, the maximum possible mutual information one could potentially measure between the prior distribution and the activated region would be $log_2(N)$ bits. Note, however, that we assigned an additional state corresponding to the fly performing a behavior outside of the significantly activated regions subsequent to the light turning on, thus making the maximal possible mutual information $log_2(N + 1)$. This additional 'zero' state is necessary to account for the possibility that the significant regions might be exhibited only in a context-dependent manner, leading to no significant phenotype when the fly is performing some behaviors at the onset of red light stimulation and leading to a phenotype if other actions are being exhibited.

# Acknowledgements

We thank Vivek Jayaraman for reagents and feedback, Jan Ache and Ugne Klibaite for discussions, Todd Laverty and members of the Janelia Fly Core for their support, and Steven Sawtelle, Igor Negroshov, Ben Arthur and Roger Rogers for help with the rig, fly bubble design and fabrication. The driver lines were developed as part of the Descending Interneuron Project Team at the Janelia Research Campus. This project was supported by the Janelia Research Campus Visiting Scientists Program and NIH GM098090.

# Additional information

## Funding

| Funder | Grant reference number | Author |
| --- | --- | --- |
| Howard Hughes Medical Institute | | Jessica Cande<br>Shigehiro Namiki<br>Wyatt Korff<br>Gwyneth M Card<br>David L Stern |
| National Institutes of Health | GM09809 | Joshua W Shaevitz<br>Gordon J Berman |

The funders had no role in study design, data collection and interpretation, or the decision to submit the work for publication.

## Author contributions

Jessica Cande, Conceptualization, Data curation, Formal analysis, Validation, Investigation, Visualization, Methodology, Writing—original draft, Writing—review and editing; Shigehiro Namiki, Resources, Data curation, Investigation; Jirui Qiu, Software, Formal analysis, Visualization; Wyatt Korff,

Conceptualization, Resources, Project administration, Writing—review and editing; Gwyneth M Card, Conceptualization, Resources, Supervision, Project administration, Writing—review and editing; Joshua W Shaevitz, Conceptualization, Supervision, Funding acquisition, Project administration, Writing—review and editing; David L Stern, Conceptualization, Supervision, Funding acquisition, Methodology, Writing—original draft, Project administration, Writing—review and editing; Gordon J Berman, Conceptualization, Data curation, Software, Formal analysis, Supervision, Validation, Investigation, Visualization, Methodology, Writing—original draft, Writing—review and editing

### Author ORCIDs
Shigehiro Namiki http://orcid.org/0000-0003-1559-799X
Wyatt Korff http://orcid.org/0000-0001-8396-1533
Gwyneth M Card http://orcid.org/0000-0002-7679-3639
David L Stern http://orcid.org/0000-0002-1847-6483
Gordon J Berman https://orcid.org/0000-0003-3588-7820

### Decision letter and Author response
Decision letter https://doi.org/10.7554/eLife.34275.029
Author response https://doi.org/10.7554/eLife.34275.030

## Additional files

### Supplementary files
• Transparent reporting form
DOI: https://doi.org/10.7554/eLife.34275.025

### Data availability
Videos including one second before until one second after activation for all flies during all treatments have been uploaded to Dryad (doi:10.5061/dryad.fr89c0c). Videos were slowed down 4X to allow easier examination.

The following dataset was generated:

| Author(s) | Year | Dataset title | Dataset URL | Database, license, and accessibility information |
|---|---|---|---|---|
| Cande J, Namiki S, Qiu J, Korff W, Card G, Shaevitz JW, Stern DL, Berman GJ | 2018 | Data from: Optogenetic dissection of descending behavioral control in Drosophila | http://dx.doi.org/10.5061/dryad.fr89c0c | Available at Dryad Digital Repository under a CC0 Public Domain Dedication |

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
