## [Decision Letter]

Thank you for submitting your article "Optogenetic dissection of descending behavioral control in *Drosophila*" for consideration by *eLife*. Your article has been evaluated by Gary Westbrook (Senior Editor) and three reviewers, one of whom is a member of our Board of Reviewing Editors. The following individual involved in review of your submission has agreed to reveal his identity: Andrew D Straw (Reviewer #2). The reviewers have discussed the reviews with one another and the Reviewing Editor has drafted this decision to help you prepare a revised submission.

Summary:

In this manuscript, Cande and colleagues leverage three new tools to dissect the role of descending neurons in controlling *Drosophila* behavior: split-Gal4 lines targeting DNs, a high-sensitivity, long-wavelength optogenetic channel, and t-SNE based behavioral analysis. They claim that activation of most DNs yield stereotyped actions, these actions are common across DNs, and the resulting action depended on previous behavioral expression. The manuscript is well written and the study is very timely given the emergence of new tools permitting the comprehensive study of DNs and behavior. Perhaps the most valuable aspect of the paper will be by providing behavioral phenotypes to use when investigating the more detailed physiology and behavioral function of the relevant circuits. Although this is a very exciting study, significant concerns were raised regarding the video data and analysis that need to be addressed.

Essential revisions:

1) The Videos 1-5 purporting to illustrate the behaviors in question are not of sufficient quality and, in some cases, rather poorly processed. They have the following deficiencies, each of which calls into question the quality of the experiments and, therefore, the validity of the subsequent behavioral analysis:

i) Dark shaded regions appear in the video data (e.g., S03 1:58).

ii) Portions of flies are lost in some video data (e.g., S03 2:11).

iii) Boxes appear in lieu of fly images (e.g., S03 3:02).

iv) Fly orientations are sometimes wrong (e.g., S04 1;00).

v) In contrast to the text asserting that animals were imaged from the dorsal surface, flies seem to be recorded while walking on the dome (the reader sees their sides or ventral surfaces).

vi) The reader cannot easily see the relevant details of leg movements. They are rather fast. Perhaps it would help to add the human annotation information on each video segment to make it clear how the authors annotated the data.

vii) It would be useful to include longer than a couple seconds of fly videos between each cut. It is too difficult to look at extended behaviors in the current form and one cannot simply pause the video because many of the behaviors are dynamic.

2) Comparisons between experimental and control animals are not provided to the extent necessary. For example, to what extent does the entropy decrease following 'high-power' red light stimulation in control animals?

3) The watershed determination of behavior space do not seem well-aligned with the underlying heat map (Figure 1—figure supplement 2). Perhaps a higher-resolution image would better serve the authors.

4) The idea of looking at entropy reduction seems to assume that a DN's role would be to elicit specific actions rather than to modulate actions. For example, entropy would not be expected to be reduced if a DN changed behavioral state without biasing which final state was chosen.

5) In Figure 2—source data 1 and elsewhere, showing the t-SNE results without the underlying video data is too far removed from the real data to be useful to readers interested in interpreting or using the data presented in this study. Please provide relevant videos or tell the reader where to look within Videos 1-5.

6) Figure 5B is perplexing. Why should the experimental animals have a higher mutual information between behavioral states only a few seconds apart? Please expand.

7) There are apparent "anticipation" responses to light-off. For example, in Figure 4C and 4D, there are density changes starting just prior to the end of the lights-on period. Does this indicate that the flies anticipated the light change of the periodic stimulus, or is this an artifact arising from analysis? Please discuss.

8) There are apparently large differences between the behaviors exhibited by experimental and control flies prior to the stimulus onset (e.g. Figure 2C, F). Is this due to the repetitive nature of the stimulus? Where does this arise? Is it present already prior to the first lights-on event? If there are such large differences in fly behavior without optogenetic activation, it does makes me wonder if the two genotypes are in different states.

9) The authors need to make their data analysis methods more accessible by providing source code. They indicate that the tSNE embedding technique can be found in the Berman et al., 2014 paper, but I think it would be appropriate to link what seems to be the relevant source code on github https://github.com/gordonberman/MotionMapper. The authors additionally indicate that the statistical classification approach here is new and likely to be useful and I agree. Other aspects are also of likely useful, such as calculating mutual information. Therefore, as stated in the *eLife* guidelines, please "Include code used for data analysis."

10) The details of the "fly bubble" are not given. Another paper (Klibaite 2017) is referred to, but the construction details are not present in that paper, nor is even the word "bubble" mentioned.

11) It would be useful to link the behavioral findings with the underlying anatomy of DNs. Are DN inputs and outputs differently represented for different behaviors? The anatomy paper proposes different sets of DNs involved in locomotion versus flight. Do DN innervations in the VNC correlate with different behaviors? Some effort to relate the anatomy to the behavior seems important.

---

## [Author Response]

Essential revisions:

1) The Videos 1-5 purporting to illustrate the behaviors in question are not of sufficient quality and, in some cases, rather poorly processed. They have the following deficiencies, each of which calls into question the quality of the experiments and, therefore, the validity of the subsequent behavioral analysis:i) Dark shaded regions appear in the video data (e.g., S03 1:58).ii) Portions of flies are lost in some video data (e.g., S03 2:11).iii) Boxes appear in lieu of fly images (e.g., S03 3:02).iv) Fly orientations are sometimes wrong (e.g., S04 1;00).v) In contrast to the text asserting that animals were imaged from the dorsal surface, flies seem to be recorded while walking on the dome (the reader sees their sides or ventral surfaces).vi) The reader cannot easily see the relevant details of leg movements. They are rather fast. Perhaps it would help to add the human annotation information on each video segment to make it clear how the authors annotated the data.vii) It would be useful to include longer than a couple seconds of fly videos between each cut. It is too difficult to look at extended behaviors in the current form and one cannot simply pause the video because many of the behaviors are dynamic.

The reviewers have identified multiple non-optimal features of the videos, including the fact that occasionally flies wandered outside the bounding box that was captured, sometimes the flies walked on the dome or were captured in the middle of a jump or recovery, and, in rare instances, the video processing pipeline resulted in some corruption of the videos. The videos shown in these supplementary figures are a randomly selected subset of video samples from each region of behavior space. Since our data set contains 700 million frames of video, it is not reasonable (nor do we think it would be unbiased) to manually filter out these “bad” frames from the large data set. Instead we chose, as the reviewers have noted, to present the data with the flaws intact so that the reader could get an honest sense for the quality of the data.

Although the human eye is quickly drawn to the rare video clips that do not seem to match the overall pattern, we do not think that these undermine the overall mapping of the 700 million analyzed frames. It is worth noting that the vast majority of video clips show a coherent pattern of behaviors. The videos showing the response to optogenetic perturbation reflect this as well. For example, none of the videos of behaviors after optogenetic activation display image processing problems, providing confidence that our statistical signals reflect real changes in behavior and not image processing artifacts.

Regarding (vi) above, note that we did not initially manually annotate any of the video frames. Instead, regions of behavior space were manually annotated after the algorithm had clustered the behaviors. The position in behavior space for each group of sampled videos is shown just prior to the appearance of the videos. We chose to present randomly selected videos from each behavior space, rather than expert-annotated “representative” movies, to honestly represent the data that is included in the analysis.

In order to help clarify evaluation and use of the data by readers, we are preparing a ~5GB collection of videos divided not by region space, as above, by descending neuron line. These videos show clips of control and experimental flies from one second before to one second after optogenetic light activation, slowed down 4X for ease of examination. This new set of videos will allow readers to perform their own evaluation of activated behaviors and to judge the quality of the videos. We will deposit the videos in a data repository, such as Dryad, or whichever repository is preferred by *eLife*.

2) Comparisons between experimental and control animals are not provided to the extent necessary. For example, to what extent does the entropy decrease following 'high-power' red light stimulation in control animals?

This is an excellent point and we have added two additional panels (C and D) to Figure 3 showing the control flies and have noted the results in the text accordingly.

3) The watershed determination of behavior space do not seem well-aligned with the underlying heat map (Figure 1—figure supplement 2). Perhaps a higher-resolution image would better serve the authors.

We have replaced the figure with a higher-resolution image.

4) The idea of looking at entropy reduction seems to assume that a DN's role would be to elicit specific actions rather than to modulate actions. For example, entropy would not be expected to be reduced if a DN changed behavioral state without biasing which final state was chosen.

We agree that if descending neuron activation caused all behaviors to change to a randomly selected other behavior, then we would not observe a drop in the entropy. A descending neuron could, in this framework, be thought of as a “reset” switch. We would not be able to detect such a reset switch as a drop in entropy.

There is another meaning of modulatory activity, where the probability of a behavior being performed is changed or where the performance of an ongoing behavior is modified slightly, and these we should be able to detect. Note that we find that activation of many of the descending neurons alters the features of an ongoing behavior, such as walking speed, and these changes were readily detected as a drop in entropy.

To clarify the specific benefits of using entropy to detect activation phenotypes, we have added clarifying text to the “Entropy of behavior space density provides a quantitative and sensitive measure of optogenetic activation phenotypes” section.

5) In Figure 2—source data 1 and elsewhere, showing the t-SNE results without the underlying video data is too far removed from the real data to be useful to readers interested in interpreting or using the data presented in this study. Please provide relevant videos or tell the reader where to look within Videos 1-5.

We agree with the reviewers that providing the corresponding video data would be of great use to the reader. As described above, we are now preparing a large set of videos that corresponds to this line-by-line analysis and that will be freely available from a data depository.

6) Figure 5B is perplexing. Why should the experimental animals have a higher mutual information between behavioral states only a few seconds apart? Please expand.

This figure panel illustrates that the majority of experimental lines display more predictable behaviors after the activation light comes on than the control lines do. We interpret this to mean that in most experimental lines the behavior elicited by descending neuron activation shows more dependence on prior state than in control flies where the descending neuron is not manipulated. In other words, control flies have more freedom to produce a diversity of behaviors after red-light activation than do experimental flies. For example, Figures 2 and 3 provide examples of control fly behavior that is not affected by turning-on the red light. We have tried to clarify this point for readers by editing the caption to Figure 5.

7) There are apparent "anticipation" responses to light-off. For example, in Figure 4C and 4D, there are density changes starting just prior to the end of the lights-on period. Does this indicate that the flies anticipated the light change of the periodic stimulus, or is this an artifact arising from analysis? Please discuss.

We believe that this rise in density is an artifact of the temporal smoothing of the behavior data introduced by the wavelet transform and does not reflect behavioral anticipation of the light stimulus. We have added text to the caption of Figure 4 to clarify this point.

As described in Berman et al., (2014), and in the Materials and methods section here, we quantify the temporal dynamics of a fly’s posture by applying a wavelet transform to the data. Wavelet transformation involves an inherent trade-off between temporal and frequency resolution. For slower behaviors (~a few Hz) especially, this leads to a fundamental uncertainty as to when precisely the behavior starts or stops. Since the observed onset is at most 1 second before the stimulus onset, our results are consistent with the hypothesis that this apparent anticipation in fact reflects temporal smoothing.

8) There are apparently large differences between the behaviors exhibited by experimental and control flies prior to the stimulus onset (e.g. Figure 2C, F). Is this due to the repetitive nature of the stimulus? Where does this arise? Is it present already prior to the first lights-on event? If there are such large differences in fly behavior without optogenetic activation, it does makes me wonder if the two genotypes are in different states.

The time “before” stimulus onset is actually the average of 30 time periods between stimulations. We found an increase in the amount of “Idle/Slow” dynamics for experimental flies in the interstitial times between stimulations. Thus, the difference between experimental and control animals in the time “before” stimulation most likely reflects this increase in the amount of Idle/Slow behavior for experimental animals in the time between stimulation. We have clarified this point in the Figure 2 legend.

It is possible that the presence of state-dependent dynamics in the experimental animals could affect the found significant regions. However, we did not find significant differences between the first stimulation period and subsequent ones. Thus, regardless of the initial state of the fly, we found robust performance of stereotyped behaviors for most experimental flies.

Finally, we should clarify that the experimental and control flies have the same genotype and differ only in whether they were fed retinal.

9) The authors need to make their data analysis methods more accessible by providing source code. They indicate that the tSNE embedding technique can be found in the Berman et al., 2014 paper, but I think it would be appropriate to link what seems to be the relevant source code on github https://github.com/gordonberman/MotionMapper. The authors additionally indicate that the statistical classification approach here is new and likely to be useful and I agree. Other aspects are also of likely useful, such as calculating mutual information. Therefore, as stated in the eLife guidelines, please "Include code used for data analysis."

We agree, and this was part of our plan upon resubmission. We have posted replication-enabling code on GitHub at https://github.com/gordonberman/Fly_Optogenetic_Analysis so that others can assess our analysis methodology.

10) The details of the "fly bubble" are not given. Another paper (Klibaite 2017) is referred to, but the construction details are not present in that paper, nor is even the word "bubble" mentioned.

We agree that we had provided insufficient details to allow replication of this apparatus. We have added additional details that we hope are sufficient to allow a reader to replicate our methods.

11) It would be useful to link the behavioral findings with the underlying anatomy of DNs. Are DN inputs and outputs differently represented for different behaviors? The anatomy paper proposes different sets of DNs involved in locomotion versus flight. Do DN innervations in the VNC correlate with different behaviors? Some effort to relate the anatomy to the behavior seems important.

We performed some of these suggested analyses, but the results were not straightforward to interpret. In some cases we see association between anatomy and behavior, but in other cases we do not. There are multiple possible causes for the lack of correlation in some cases and we believe that this problem requires significant further analysis before it would be helpful to readers. This is a focus of our future efforts.